# Photo-Elicited Conversations about Therapy Dogs as a Tool for Engagement and Communication in Dementia Care: A Case Study

**DOI:** 10.3390/ani9100820

**Published:** 2019-10-17

**Authors:** Lena Nordgren, Margareta Asp

**Affiliations:** 1Centre for Clinical Research Sörmland, Uppsala University, 631 88 Eskilstuna, Sweden; 2Department of Public Health and Caring Sciences, Uppsala University, 752 36 Uppsala, Sweden; 3School of Health, Care and Social Welfare, Mälardalen University, 722 20 Västerås, Sweden; margareta.asp@mdh.se

**Keywords:** animal-assisted therapy, dementia, case reports, observation, phenomenology, hermeneutics, qualitative research

## Abstract

**Simple Summary:**

Animal-assisted therapy is a meaningful and pleasant activity that can evoke feelings of joy and pleasure in people with dementia. In this study, the researchers wanted to find out whether photos of a person with dementia interacting with a therapy dog could be used to promote positive feelings and memories for the person in a similar way. Repeated video observations of photo-elicited conversations between a woman with dementia and a dog handler/assistant nurse were used to collect data. The results indicate that photo-elicited conversations can be used to talk about feelings and memories. With the help of contemporary digital technology, photobooks are relatively inexpensive and uncomplicated to make. They are easy to handle for persons with dementia and for family members or nursing staff. In addition, photobooks can be kept readily available for the person with dementia so they can look at them without assistance. To increase a sense of self in people with dementia, the photos should portray them in recent positive situations, for example, when interacting with an animal such as a therapy dog.

**Abstract:**

Understanding the inner life of people with dementia can be challenging and there is a need for new and different approaches. Previous research shows that people with dementia can experience emotions such as harmony, closeness, and joy as well as sadness and concern when interacting with a therapy dog. Simultaneously, memories of past episodes are brought back to life when the person interacts with the dog. This raises questions about whether photos of interaction with a dog can evoke memories or support people with dementia in communicating emotions in a corresponding way. The aim of this study was to explore photo-elicited conversations as a tool for engagement and communication in dementia care. Repeated video observations of photo-elicited conversations between a woman with dementia and a dog handler/assistant nurse were used to collect data. The video recordings were analyzed with a phenomenological hermeneutical method. The role of photo-elicited conversations as a tool for engagement and communication in dementia care is that the conversations can help the person with dementia to feel a sense of being situated and recall feelings of liveliness and belongingness, and thus supporting the person’s sense of self. The results can be used to deepen nursing staff’s understanding of using photo-elicited conversations in dementia care.

## 1. Introduction

Animal-assisted therapy (AAT) is a therapeutic intervention that is goal-oriented, planned, and structured. In Sweden, it is usually prescribed by qualified healthcare professionals and delivered by formally trained, certified professionals with expertise within the field of his or her practice and with a trained pet, usually a dog. The progress of the intervention is measured and documented in, for example, medical records [1]. In AAT, therefore, trained pets participate in structured programs that focus on the improvement of specific outcomes in people.

About 50 million people around the world suffer from dementia [2]. In 30 years, this number will have increased to 150 million people. Dementia is not a part of normal ageing, even though it is elderly people who are mostly affected, and it is not one single disease but a syndrome caused by different underlying conditions. The most prominent symptoms of dementia are memory problems and declined cognitive functions. This leads to reduced capacity to perform daily activities. Later, these abilities are further impaired due to physical incapacities. Many people with dementia also suffer from behavioral and/or psychological symptoms [2]. The person’s reduced functional capacity means a decreased ability to express thoughts, needs, and emotions. In turn, this reduces their possibility to communicate and interact with other people. Consequently, people with dementia are vulnerable and often experience reduced well-being and poor quality of life [3]. 

With the intention of improving well-being and quality of life, communication and the effects of communication have received increasing attention in research and in dementia care [4]. Communication and interaction with other people is important for our perception of personhood and self [5] because this promotes experiences of being socially involved. A person’s sense of self has its origin in the past and is reconstructed in the present. A positive sense of self is important for well-being, but cooperation from others is required [6]. Activities that involve communication and interactivity can, therefore, strengthen a person’s identity and sense of self.

Animal-assisted therapy seems to have beneficial mental and physical effects on mood, reactivity, and overall quality of life in people with dementia [7,8]. In a recent review of AAT for people with dementia, positive effects were identified regarding communication skills, verbal and body language and interaction, well-being, self-esteem, and mental attitude [7]. The authors suggest that AAT increases the desire to participate in activities and in groups. Animal-assisted therapy can, therefore, improve quality of life, mood, motivation, and social behavior [7]. Another recent systematic review [8] concluded that there seems to be a general consensus regarding the efficacy of animal-assisted intervention (AAI) in improving social functioning as well as moderating effects on the behavioral and psychological symptoms of dementia. However, based on existing research findings, it is difficult to draw firm conclusions [7,8] and more research about AAI and AAT in dementia care is needed [9].

Most studies about AAI in dementia care are quantitative and there are not many qualitative studies, especially not from the perspective of persons with dementia. A recent review [10] over qualitative evidence for the possible mechanisms for the effectiveness of AAI showed that the effectiveness of AAI is related to the contact with the animal by, for example, bonding or interaction rather than to the animal’s appearance or cuteness. According to findings of another qualitative study, people with Alzheimer’s disease are able to express empathy and love by showing joy and tenderness for a therapy dog [11]. The authors’ interpretation was that, in the presence of a therapy dog, people with Alzheimer’s disease can, for a while, feel important, needed, and meaningful to someone, i.e., the dog. Findings in another similar study indicated that people with Alzheimer’s disease were able to perceive themselves as “whole” human beings in meetings with therapy dogs and that they were able to connect with their inner feelings. This evoked awareness of their past and present life [11]. 

Digital photo diaries have been suggested as a tool for supporting communication between people with Alzheimer’s disease and their families [12]. The photos can stimulate the ability to talk about recent and past events, and the possibility to share memories can increase the person’s participation in everyday life; thus, supporting the person’s sense of self. In addition, digital diaries can be used by home care nurses who provide person-centered care [13]. However, more research is needed to gain a better understanding of how to use photos in dementia care.

Dementia is often signified by a reduced ability to express emotions or memories. The use of visual materials in interviews can aid researchers in collecting data about what can be seen in the photo, but visual materials can also allow the researcher to elicit the participant’s abstract perceptions about the world depicted. Questions that come to mind concern what happens when people with dementia are asked to talk about pictures of themselves involved in a specific recent event. What do they express when they look at the photos? What memories or emotions are evoked? How do they react to the pictures? The aim of this study was to explore the role of photo-elicited conversations as a tool for engagement and communication in dementia care.

## 2. Materials and Methods 

A qualitative research design was applied. The setting was a Swedish nursing home for residents with dementia. Repeated video recordings of photo-elicited conversations with a woman with dementia were used to collect data. The woman is referred to in this paper as “Mrs. Anderson”; see Box 1 for more information about Mrs. Anderson. The recordings were analyzed with a phenomenological hermeneutical approach [14,15]. The study was conducted in accordance with the Declaration of Helsinki and the Regional Ethical Review Board in Uppsala, Sweden approved the study before it started (2017/081). The participant, Mrs. Anderson, was chosen because she was the first of consecutive participants in a larger study that will be reported elsewhere. Mrs. Anderson and her husband gave informed consent to participate in the study in November 2017.

Box 1Portrayal of Mrs. Anderson.Mrs. Anderson was an 82-year-old woman with vascular dementia who became a nursing home resident in August 2017. As well as dementia, she also had some cardiovascular diseases. Mrs. Anderson had been married to the same man for more than 50 years and they had two sons. Mr. and Mrs. Anderson owned a house and a dog. She had been an active and social person all her life and was still very communicative.An occupational therapist prescribed dog-assisted therapy for Mrs. Anderson in September 2017 as she appeared increasingly depressed and passive. She spent a lot of time lying in bed and did not get up or get dressed. Mrs. Anderson told the dog handler she was only in bed for a nap, but according to the nursing staff, she spent most of her time lying in bed wearing her nightdress. Furthermore, Mrs. Anderson displayed behavioral and psychological symptoms of dementia, such as physical and verbal aggression and delusions. However, when the dog handler and the dog showed up for AAT, she stood up and got dressed, and participated actively and enthusiastically in the activity. The same behavior was seen in connection with the video recordings. Mrs. Anderson would be lying on her bed when the research team arrived, but when she realized they would be talking about the dog, she immediately got up and participated in the conversations with great enthusiasm.The dog-assisted therapy aimed to motivate and strengthen Mrs. Anderson in her daily activities. The intervention lasted for 6 weeks and included ten sessions each lasting from about 45 min to 1 h. The therapy sessions included activities such as playing games or taking the dog for walks outdoors. During some sessions, the dog handler brought a camera and took photos of Mrs. Anderson and the dog. The photos were kept in a specially designed photobook that was accessible for Mrs. Anderson in her room. The photobook was later used in the present study.To evaluate the effects of the AAT and in accordance with the nursing home’s standard procedure, nursing staff assessed Mrs. Anderson’s quality of life with The Quality of Life in Late-Stage Dementia scale (QUALID). QUALID is a questionnaire specifically developed to measure quality of life in later stages of dementia [16,17]. The questionnaire consists of 11 statements about positive and negative behaviors. The responses are graded on occurrence during the previous week on a five-point scale. The points are summed up to a total of between 11 (best quality of life) and 55 (worst quality of life). During the period of AAT, Mrs. Anderson’s quality of life was relatively stable. Before AAT, her QUALID score was 34. One week after the completed intervention, the QUALID score was 37. One month after the completed intervention, the QUALID score was once again 34.An occupational therapist completed the Mini-Mental State Examination (MMSE) before the first video recording and one week after the last video recording [18]. Before, Mrs. Anderson had mild dementia as indicated by an MMSE of 27 and after, the MMSE had decreased to 23 (mild dementia).

### 2.1. Data Collection

Photo elicitation is a technique whereby photos are included in an interview situation [19,20]. Photos can enrich the research process and serve as door openers as they often evoke spontaneous, vivid, unpredictable, and personally meaningful responses from study participants [19,21]. Moreover, photos can prompt memories and information that would not be discussed in a conventional interview [22], thereby moving the interview beyond the verbal [21]. If the interview is successful, the researcher’s and the participant’s understandings of the meaning of the image increases [20].

In this study, a specially designed photobook was used that contained photos of Mrs. Anderson during different sessions with the therapy dog; the same photobook was used in all video recordings. The book was kept in Mrs. Anderson’s room and she could access the book at any time.

Mrs. Anderson, the dog handler, and one member of the research team met for video recordings on four different occasions in November and December 2017, and in February and May 2018. The video recordings were conducted in the afternoon and lasted between 14 and 38 min. Sadly, the dog passed away between the second and third video recordings, which Mrs. Anderson was told about during the third video recording. Due to technical problems, the fourth video recording was unusable.

Before each video recording, the dog handler and the researcher approached Mrs. Anderson and asked whether she felt like talking about the photos, to which she agreed. The dog handler and Mrs. Anderson sat down in Mrs. Anderson’s room while the researcher stood 1 to 3 meters away, behind the camera. Mrs. Anderson was aware of the researcher and the video camera. No one else entered the room during the video recordings. The dog handler asked Mrs. Anderson if she could talk about what she saw in the photos. Mrs. Anderson talked about the photos and about whatever came into her mind. At times, she would turn to the researcher to comment on the pictures or ask questions; there had been no prior relationship between Mrs. Anderson and the researcher.

In addition to the video recordings, a registered nurse at the nursing home provided the researchers with relevant information about Mrs. Anderson’s health status and dementia symptoms. The researchers, therefore, did not personally access Mrs. Anderson’s medical chart.

### 2.2. Data Analysis

The video recordings were transcribed into text. The transcripts were analyzed with a phenomenological hermeneutical method as described by Lindseth and Norberg [15]. The process implies a dialectical movement between the whole and the parts of the data and involves three hermeneutical arcs.

The first arc of interpretation is the ‘naïve understanding’ that evolves while observing all recordings and reading all interviews repeatedly, with openness, in order to understand the data material as a whole and the meaning of the phenomenon [15]. Both authors watched the three video recordings together once. Important sequences were noted and transcribed. Both verbal and non-verbal communication was included. The content was discussed and reflected on until agreement was reached over a descriptive initial naïve understanding.

The second arc of interpretation, the ‘structural analysis’, focuses on explaining the phenomenon [15]. This process is characterized by a reflective distance to condense meaning units in the text and create themes and sub-themes. The themes were validated back and forth in relation to the naïve understanding, which also developed the description of the naïve understanding.

The third arc of interpretation, the ‘comprehensive understanding’, is a critical synthesis of the evolving results [15]. To gain a deeper understanding of the phenomenon, the critical synthesis is interpreted in relation to other texts, such as previous research findings or philosophical ideas. Thus, the whole interpretation process means ‘a move from understanding to explaining and then a move from explanation to comprehension’ [14].

## 3. Results

The results are presented in accordance with the three arcs of interpretation. First, the naïve understanding is described followed by the results from the structural analysis. Finally, the comprehensive understanding is included in the discussion.

### 3.1. Naïve Understanding

During the photo-elicited conversations, Mrs. Anderson tried to relocate herself in the environment, that is, she situated herself in time and space. She recognized the dog, and it appeared that the pictures supported her memory, which means, she recalled the sessions with the dog. She expressed joy, pleasure, well-being, love, and affection towards the dog. During the conversations, she also communicated longing and desire. She described that the dog listened genuinely to her and that their contact was deep and sincere. The photos were used as consolation when she was lonely. A chain of memories started, and Mrs. Anderson communicated memories of loved and lost significant others.

### 3.2. Structural Analysis

The structural analysis resulted in one main theme, three themes, and six sub-themes (Table 1).

**Main theme:***Recalling the past and recapturing the present.* The use of photo-elicited conversations as a tool for communication in dementia care means dwelling on memories, recalling the past but also recapturing the present. The conversation alternates between a need to recognize oneself and the environment and talking about childhood or relationships with other people throughout the lifespan. The photos evoke feelings of joy and affection, sadness, loneliness, and belongingness.

**Theme** **1.**
*Sense of being situated means that the photo-elicited conversations supported Mrs. Anderson in identifying herself in a context or time that made sense. This implies that photo-elicited conversations can evoke a sense of being situated.*


*Recognizing oneself.* When Mrs. Anderson began to look at the photos, she was not able to recognize herself. She asked who the woman in the photos is. When the nurse explained that it was Mrs. Anderson in the photos, she took another look at the photos and then recognized herself. She pointed out that her hair used to be different and she said that this was the reason why she did not recognize herself at first.

*Yes …it is me…I didn’t see that it was me. Mm my hair is light, but it has become lighter*.

Mrs. Anderson pointed at the photos, repeatedly touched her hair, and eventually accepted that the woman in the photographs was her.

*Reacquainting oneself with the environment*. Mrs. Anderson reacquainted herself with the environment in the photos with energetic explanations. She used verbal explanations and gestures to illustrate the nature, the rocks, and the footpath illustrated in the photos.


*This was when we took long walks…and you…then it starts…it goes downhill and then the hill starts here. And then behind it is the house here…*


She also looked out of the window and re-orientated herself in the room as well as in the nursing home. When she looked at the photos, she believed that they were taken during the summer. The nurse explained the pictures were taken last autumn. Then Mrs. Anderson noticed that she was wearing a knitted cap, scarf, and gloves in the photos and she agreed that the photos must have been taken in the autumn.

When Mrs. Anderson talked about the environment, she also recognized her own home. She told the nurse that she lived here now [*in the nursing home*] but that her husband did not. She said it was difficult to be separated from her husband because she was not used to that.

**Theme** **2.**
*Recalling liveliness and losses means that photo-elicited conversations provide opportunities to talk about both joyful and difficult memories. The photos serve as a tool for talking about strong feelings in moments of liveliness, love, and compassion. At the same time, the photos bring feelings of regret and memories of losses throughout life.*


*Experiencing liveliness.* Mrs. Anderson recalled experiences of joy and the beauty of nature from the meetings with the dog. When she talked about these moments, she expressed the same feelings and liveliness. These feelings brought about engagement and vitality.

*Yes, that was then, yes. And Milou, yes…yes, just think how much fun it was when we were out in the forest. And when I see this, then I’ll be …I’ll be like… when I see this… this picture, I just feel, I just want to run right out there when I see this. Yes, oh it looks like such fun, mm. So wonderful*.

She explained that the place where the photos were taken was very beautiful and she also understood the beauty in the picture.


*What a wonderful picture. Beautiful. You don’t think it can be…with the stones…stones that are grey and slightly green and all this…*


These experiences of beauty evoke feelings of joy and vitality.

*Experiencing loss.* During the photo-elicited conversations, Mrs. Anderson told that she had looked at the photos several times. The pictures were a substitute for nearness to the dog and Mrs. Anderson had the pictures under her pillow so that she could look at them in the evenings.

*Yes, I have looked and yesterday evening when I lay down…the last thing I did before I went to sleep was to have this (the photobook). Oh… I’ve looked at this so much in the evenings*.

She longed for the dog and missed their walks. While she looked at the photos, she explained that she wanted to go outside, but regretfully she was not allowed.

*But I miss you so much because it’s not so easy to go out and just take off. If… I’m … I’m not allowed to go out alone you know. That’s what’s so sad*.

During the third video session, the nurse had to tell Mrs. Anderson that the dog passed away. Even before the nurse told her, Mrs. Anderson realized this from the look on the nurse’s face.

*He was so unusual. I’ve cried when I’ve thought about Milou. I have looked at Milou and the tears have flowed. I didn’t know that he had died*.

The dog’s death raised concerns about other losses throughout her life of dogs, people, and a dear brother.


*No, I feel so, I don’t understand …I’m going to …yes (crying) why can’t they stay when you… why… why does it need to happen this way… when you like them so much. It’s so…I lost one of my brothers, I have three… we have three brothers (crying)…*


When the emotions became too strong, Mrs. Anderson dealt with her losses by suddenly changing the subject of the conversation to avoid the difficulties.

**Theme** **3.**
*Recalling a sense of belonging means that photo-elicited conversations evoke awareness of the relationship with the dog and with other people. The conversation strengthened Mrs. Anderson’s experience of belonging with other living beings.*


*The relationship with the dog*. Mrs. Anderson described her connection with the dog. She recalled her times with the dog as happy, incredible, or remarkable. The dog was described as good, beautiful, cuddly, soft, sweet, kind, and special. She perceived that the dog really listened to her with attention and looked into her eyes.

*I don’t think I’ve ever met a dog that has been like him… and here… he listened to… he listened to what… I wasn’t his owner but he listened when I was talking and that shows/…/yes… he looks … I look… we look each other in the eyes. Yes… he was so lovely and kind /…/ It meant so much to me here in my heart. He really listened. And that is what is so fantastic, that you can have a dog that listens*.

She moved her body and used gestures to illustrate how the dog was leaning his head and stretched out as he listened to her. He was a true connoisseur of humans, she said. Mrs. Anderson used to hug the dog and kiss his nose. She explained that she felt an incredible contact with the dog that went straight to her heart. The feeling was completely different from the sense of being alone.

*The relationship with the nurse and with relatives*. The photo-elicited conversation involves a relationship with the nurse. They sit close together, face to face, and sometimes they touch to emphasize what they mean. The nurse listened attentively to what Mrs. Anderson said and followed her reflections and associations while posing comments and questions.

During the third video session, when the nurse told with sorrow that the dog has died, Mrs. Anderson expresses empathy and compared the nurse’s sorrow with how she would feel in the same situation.

*If you like someone why does it have to be like this? How is it going to be for you now? No I understand you. I would cry terribly*.

The photo-elicited conversation aroused associations with relatives in Mrs. Anderson’s life. Sometimes she could not remember if they were dead or alive. She recalled good memories of their relationships.

## 4. Comprehensive Understanding and Discussion

The use of photo-elicited conversations as a tool for communication in dementia care means dwelling on memories, recalling the past but also recapturing the present. This means being situated in temporality and spatiality, which can be difficult for persons with dementia. The present results are interpreted in relation to the existential phenomenological tradition as described by the French philosopher Merleau-Ponty [23].

*Temporality* refers to time as it is humanly experienced and gives every experience a temporal meaning [23]. For Mrs. Anderson, temporality was evident in her longing to meet the dog and her feeling that it had been a long time since they met. These temporal meanings relate to our everyday lives and what it is to be human.

*Spatiality* refers to the environment, a world of places and things that have meaning in our lives [23]. Things are regarded as close or distant in terms of their significance in our daily lives. Mrs. Anderson described the dog as being very close to her, while the presence of the longed-for dog felt close in his absence. Spatiality also refers to the beautiful pictures and the nature that Mrs. Anderson recalls and talks about. She appreciated the big stones covered with moss and the beautiful surroundings. Perceiving such impressions of nature is spontaneous and arouses curiosity. Kaplan and Kaplan [24] define this condition as “soft fascinations” and this kind of attention regenerates and restores individuals and their powers. This is an example of how spatiality can affect individuals’ health.

Intersubjectivity refers to how humans live in the world with other living beings. From a phenomenological perspective this means that “this world is not just open to other human beings but also to animals who dwell in it after their own fashion; they co-exist in the world” [25] (p. 70). Mrs. Anderson expressed how being out in nature created vitality. This can be interpreted in relation to Searle’s theory [26] that says that nature sparks creative and rehabilitative processes for human beings. Mrs. Anderson particularly recognized the stones and she always found them at the same spot. According to Searles [26], our relationships with stones and inanimate objects are the simplest relationships. The most complicated relationships are those with other people, and our relationships with plants and animals fall in-between. It is possible that this theory explains why Mrs. Anderson made such good contact with the dog and could communicate with him.

The conversations also meant that Mrs. Anderson recalled experiences of belongingness and memories of losses throughout her life. Previously, the concept of “at-homeness” has been used to metaphorically describe experiences of being at home in spite of disease and illness [27]. The meanings of at-homeness for elderly people with severe illness are temporally and spatially shaped when the ill person recognizes her-/himself and is known or seen by others [28]. At-homeness can be shaped either in a process or momentarily [28]. To use a photobook that includes photos of the person with dementia engaging in meaningful activities, such as spending time with a therapy dog, could presumably mean that feelings and memories are revived so that the person with dementia will feel a sense of being situated. Thus, the person may, for a while, experience at-homeness.

Within healthcare, and in communication with a person with dementia, a lifeworld perspective that includes the person’s experiences of temporality, spatiality, and intersubjectivity can improve the understanding of how the person perceives his or her life. Care can then be planned based on this understanding [29] and, in turn, this can support the person’s sense of self.

### 4.1. Clinical Implications

It is well known that people with dementia spend much of their time in nursing homes being unoccupied [30]. This can exclude them from structured activities or social interactions [31,32], thus reducing their quality of life [31]. Understimulation can also negatively affect different problematic behaviors [30]. In contrast, studies indicate that engagement in individualized meaningful activities can positively affect behavioral symptoms that would otherwise exclude people with dementia from structured activities [33,34]. Meaningful activities can increase feelings of pleasure or positive gestures, as well as reduce anxiety and anger, negative verbalizations, or other negative nonverbal behaviors [35]. Moreover, people with dementia can be effectively engaged by different stimuli; the most engaging stimulus is so-called one-on-one socializing [30]. In turn, personalized self-identity stimuli, for example, looking at family photos, can evoke engagement in people with dementia [30]. In the present study, no improvement in Mrs. Anderson’s quality of life was observed. Still, the photo-elicited conversations can be understood as a one-on-one socializing meaningful activity, which, in part, may explain Mrs. Anderson’s engagement in the conversations about the photos of herself with the dog.

The ability to talk about recent events can vary between people with dementia [13]. In this study, the main focus was on Mrs. Anderson’s relationship with the dog. When she looked at the pictures, she vibrantly and with affection described the emotions that she felt. In addition, she was able to narrate memories in detail and with compassion from her previous meetings with the dog. Furthermore, the images started memory chains and she told the dog handler and the researcher about her youth, her father, her brothers, other dogs, her husband, and her sons. Karlsson and co-workers [13] found that their study participants attempted to understand themselves and their sense of self by using images of themselves in different situations, and different pictures could bring different associations. Hence, it seems reasonable to assume that if Mrs. Anderson had not previously met the dog, and if she had not had a strong and affectionate bond with the dog, the images would probably not have evoked such strong emotions or memories. This means that if photobooks are used in clinical settings, it is important to choose the images with caution and base them on the person’s individual preferences and interests.

Previous studies have used different technical/digital aids such as videos or iPad apps [36,37]. One problem with technology, though, is that it can be difficult for a person with dementia to use on their own, without any kind of assistance. A photobook similar to the one that was used in this study is cheap and easy to handle, even for a person with dementia. All that is needed is a mobile phone with a camera and someone who is willing to take the time to create the photobook using a simple online service. The book can be kept easily accessible for the person with dementia so that he or she can pick it up at any time and look at the images on their own. Since the photobook is easily accessible, it is also easy for families or nursing staff to pick up and use for distraction or for meaningful conversations with the person with dementia.

Mrs. Anderson and the dog handler had a prior caring and affectionate relationship from the previous AAT. On the basis of the present study, it is not possible to conclude whether the conversations were affected by that prior relationship or not. In addition, the dog handler was an experienced assistant nurse who worked daily in line with the principles of person-centered care. The dog handler/nurse was careful to recognize Mrs. Anderson and Mrs. Anderson’s emotions, and the conversations were characterized not only by affection and care, but also by respect and trust, which is consistent with person-centered care [38]. Thus, it is reasonable to assume that both the photos and the prior relationship with the dog handler affected Mrs. Anderson’s engagement in the photo-elicited conversation. This needs to be further studied.

In turn, it is not possible to determine the effects of the photo-elicited conversations on Mrs. Anderson’s ability to communicate. Still, for a moment, Mrs. Anderson’s apathy was reduced and the conversations were considered to possibly bring a sense of meaningfulness. It can be noted, however, that Mrs. Anderson’s verbal communication seemed appropriate in the situation. Previous studies about AAI have shown similar results [39,40,41]. Being engaged in communication and in activities is relevant for our sense of self since it leads to social involvement and social inclusion. It is reasonable to assume that if nursing homes provide activities that emanate from the persons themselves, this can support them in maintaining a sense of self even in more severe stages of the disease.

Furthermore, it is essential to consider the time of day when a structured activity in dementia care is organized. In the present study, the photo-elicited conversations were conducted in the afternoons. People with dementia have a lower stress threshold and their stress levels seem to increase throughout the day [32]. For that reason, activating activities, for example, AAT, are appropriate to schedule in the morning while activities that relieve anxiety by providing soothing environmental stimulation, for instance photo-elicited conversations, are more appropriate in the afternoon. However, this requires a deliberate planning process [32].

More research is needed in order to determine the amount of time that can elapse between the actual event (depicted in the photos) and the photo-elicited conversations. Six months after the AAT, i.e., at the time of the fourth video recording (which unfortunately was unusable for analysis), the memories from the meetings with the dog were still available to Mrs. Anderson and she talked vividly about them. Other factors beside the time aspect probably matter, for example, the severity of the disease. However, photo-elicited conversations are possibly an appropriate activity even for people with severe dementia.

### 4.2. Limitations

To support an in-depth exploration of photo-elicited conversations as a tool for engagement and communication in dementia care, the present study used a single-case study research design with an observational method. Case studies are appropriate when the researchers want to explore a phenomenon in its context or study the behavior of a single individual [42]. In turn, observational methods are used in case studies to illuminate a case from all sides [42]. Case studies can be analyzed using a variety of methods. In order to obtain knowledge of the essential meaning of the phenomenon in its context, a phenomenological hermeneutical method seemed appropriate [15].

The results of this study cannot be generalized but they can be used to illuminate other, related cases in forthcoming studies [42]. In a small single study like this, there is not a “single fundamental truth because the whole truth can never be fully understood” [15]. Instead, the researchers search for possible meanings. In order to achieve trustworthiness, detailed descriptions of the case and the research process are provided. Quotations are used to illuminate the findings of the structural analysis. The second author conducted the video recordings and, by watching the recordings together, both authors were able to become immersed in the situation, that is, become aware of the research context and gain insight into the setting. This “prolonged engagement” helped the researchers to deepen their evolving understanding [42]. The whole research process—from planning to reporting the results—has been characterized by discussions, reflexivity, and the authors’ self-critical stance [42].

## 5. Conclusions

The use of photo-elicited conversations as a tool for communication in dementia care involves dwelling on memories, recalling the past but also recapturing the present. The conversations imply that a person with dementia can have a feeling of being situated and of liveliness. In view of this, conversations elicited by photos of people with dementia involved in animal-assisted therapy seem to have the capacity to create engagement and meaningfulness for them. In line with principles for person-centered care, deliberate planning and individualized activities for people with dementia mean that a person’s remaining abilities and resources can be supported. Thus, for people with dementia, AAT and photo-elicited conversations about the precious animal offer meaningful, one-on-one activities that can increase their quality of life and their sense of self. More research is needed in order to further investigate the role and the place of photo-elicited conversations and animal-assisted therapy in dementia care.

## Figures and Tables

**Table 1 animals-09-00820-t001:** Main theme, themes, and sub-themes from the structural analysis.

Main Theme	Theme	Sub-Theme
Recalling the past and recapturing the present	Sense of being situated	Recognizing oneself
		Reacquainting oneself with the environment
	Recalling liveliness and losses	Experiencing liveliness
		Experiencing loss
	Recalling a sense of belonging	The relationship with the dog
		The relationship with the nurse and with relatives

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
