# Peer review of "Photo-Elicited Conversations about Therapy Dogs as a Tool for Engagement and Communication in Dementia Care: A Case Study"

_animals, 2019, doi:10.3390/ani9100820_

Round 1

Reviewer 1 Report

The case report highlights a really interesting topic. However, some consideration has been reported. I suggest, in order to improve the general quality of the manuscript, to avoid using colloquial words (es line 210-216). This is the main problem of the text.

Line 50 : It is not clear what does it mean {IAHAIO, 2018 #160}. Is the type of medical records?

Line 52: I don't understand this reference and the following ones {, 2019 #137}.

Line 53 : reference is missing "number will have increased to 150 million people"

Line 61: why the reference is different now

Line 185: reference is not clear

LIne 382: references are mixed up

Author Response

We really appreciate the time and efforts you have put in the review of our manuscript! Thank you!

We are sorry about the references. Our EndNote program seems to have gone wild... We have now corrected all references throughout the paper and we hope you find the corrections agreeable.

We have also changed the wording in the Naive Understanding. It now reads:

During the photo-elicited conversations, Mrs. Anderson tries to relocate herself in the environment, that is, she situates herself in time and space. She recognizes the dog, and it appears that the pictures support her memory, which means she recalls the sessions with the dog. She expresses joy, pleasure, well-being, love, and affection towards the dog. During the conversations, she also communicates longing and desire. She describes that the dog listened genuinely to her and that their contact was deep and sincere. The photos are used as consolation when she is lonely. A chain of memories starts, and Mrs. Anderson communicates memories of loved and lost significant others.

We hope you will find the revision more enjoyable!

Thanks again!

Reviewer 2 Report

General comments: The objective of this study was to test the use of photo-elicited conversations as a tool to improve the health of dementia patients. The authors present a case study to support this method.

The study is very interesting and informative, and can be very useful for experts working with people with dementia. The paper is very well written and easy to read.

I have no particular suggestion, except for the formatting of references in the text that doesn’t follow the guidelines of the journal:

In the text, reference numbers should be placed in square brackets [ ], and placed before the punctuation; for example [1], [1–3] or [1,3]. For embedded citations in the text with pagination, use both parentheses and brackets to indicate the reference number and page numbers; for example [5] (p. 10), or [6] (pp. 101–105).

Author Response

We really appreciate the time and efforts you have put in the review of our manuscript! Thank you!

We are sorry about the references. Our EndNote program seems to have gone wild... We have now corrected all references throughout the paper and we hope you find the corrections agreeable.

Thanks again!

Round 2

Reviewer 1 Report

Thanks for the accurate revision.